# Effectiveness of Abdominal Ultrasonography for Improving the Prognosis of Pancreatic Cancer during Medical Checkup: A Single Center Retrospective Analysis

**DOI:** 10.3390/diagnostics12122913

**Published:** 2022-11-23

**Authors:** Atsushi Yamaguchi, Naohiro Kato, Shuhei Sugata, Takuro Hamada, Nao Furuya, Takeshi Mizumoto, Yuzuru Tamaru, Ryusaku Kusunoki, Toshio Kuwai, Hirotaka Kouno, Naoyuki Toyota, Takeshi Sudo, Kazuya Kuraoka, Hiroshi Kohno

**Affiliations:** 1Department of Gastroenterology, Kure Medical Center and Chugoku Cancer Center, Kure 737-0023, Japan; 2Department of Radiology, Kure Medical Center and Chugoku Cancer Center, Kure 737-0023, Japan; 3Department of Surgery, Kure Medical Center and Chugoku Cancer Center, Kure 737-0023, Japan; 4Department of Pathology, Kure Medical Center and Chugoku Cancer Center, Kure 737-0023, Japan

**Keywords:** pancreatic cancer (PC), abdominal ultrasonography (US), surveillance, prognosis, medical checkup, 5-year survival, cancer screening

## Abstract

Recent advancements in surgical and anti-cancer therapies have provided significant hope of long survival in patients with pancreatic cancer (PC). To realize this hope, routine medical checkups of asymptomatic people should be performed to identify operable PCs. In this study, we evaluated the efficacy of medical checkups using abdominal ultrasonography (US). We retrospectively analyzed 374 patients with PC at our institute between 2010 and 2021. We divided these patients into several groups according to the diagnostic approach and compared their background and prognosis. These groups comprised PCs diagnosed through (a) symptoms, 242 cases; (b) US during medical checkup for asymptomatic individuals, 17; and other means. Of the 374 patients, 192 were men (51.3%), and the median age was 74 years (34–105). Tumors were located in the pancreatic tail in 67 patients (17.9%). Excision ratio and 5-year survival rate were significantly better in group (b) than in (a) (58.8% vs. 23.1%, *p* < 0.01 and 42.2% vs. 9.4%, *p* < 0.001, respectively). The prognosis of patients diagnosed using US during medical checkup was better than that of patients identified through symptomatic presentation of PC. US for asymptomatic individuals with PC might be one of the useful modalities for promoting better prognosis of PCs.

## 1. Introduction

Pancreatic cancer (PC) is the worst prognostic cancer, and its 5-year survival rate (5-SR) is approximately 7.1% and 10% in Japan and the United States, respectively [1,2]. It is believed that surgical intervention at an early stage improves chances of survival and improve the prognosis of patients with PC [1,2]. To achieve such positive results, we need to identify asymptomatic patients with PC. Recently, to find earlier stage PCs, attention has been paid to patients with new-onset or rapid worsening of diabetes mellitus and individuals with intraductal papillary mucinous neoplasm (IPMN) and family history (FH) of PC [3]. Nevertheless, most patients with PC present with symptoms such as jaundice, abdominal pain, appetite loss, etc. These results might be due to the fact that most patients with PC do not come from screening for IPMN and FH of PC, although these surveillances are important means for finding PCs [4,5,6,7]. In the United States, screening for PC is not currently recommended due to various reasons [8]. One reason is that there is no evidence of cancer screening improving the disease-specific morbidity or mortality of PC [8]. In addition, the Ministry of Health and Welfare of Japan do not recommend pancreatic cancer screening. In this study, we analyzed the prognosis of patients with PC who were divided into eight groups according to how PC was diagnosed and evaluated the usefulness of abdominal ultrasonography (US) during medical checkups of asymptomatic individuals.

## 2. Materials and Methods

### 2.1. Patients 

This retrospective study included 374 patients diagnosed with PC between April 2010 and June 2021 at the National Hospital Organization Kure Medical Center and Chugoku Cancer Center (Kure city, Hiroshima prefecture, Japan). The types of PC we included in this study were pancreatic ductal adenocarcinoma (PDAC), intraductal papillary mucinous neoplasm (IPMN) with high-grade dysplasia, and IPMN associated with invasive adenocarcinoma. We excluded patients with neuroendocrine neoplasm, solid pseudo-papillary neoplasm, acinar cell carcinoma, mucinous cystic neoplasm, and pancreatic metastasis from other cancers. This study was performed in accordance with the Declaration of Helsinki and was approved by our ethics committee (No. 2022-24). Patients were not required to provide informed consent to the study because the analysis was performed using anonymous clinical data. For disclosure, the details of the study are posted on some walls in the National Hospital Organization Kure Medical Center and Chugoku Cancer Center.

### 2.2. Initial Diagnosis and Follow-Up

The height and body weight of the participants were assessed and any history of comorbidities (especially, diabetes mellitus), malignancies, alcohol intake or smoking and FH of PC were recorded. Patients underwent blood tests, abdominal contrast enhanced computed tomography scans (CE-CT), magnetic resonance cholangiopancreatography (MRCP), and endoscopic ultrasonography (EUS) during their first visit to our hospital. Further, they underwent fine-needle aspirations using EUS and/or pancreatic juice cytology using endoscopic retrograde pancreatography (ERP). Presently, we use positron emission tomography (PET) and hepatobiliary magnetic resonance imaging with gadoxate disodium for detecting distant metastasis. IPMN-derived carcinoma was differentiated from concomitant PDAC in IPMN based on an assessment of the continuity of the carcinoma and IPMN using imaging studies or pathological examinations. For diagnosis, we used surgical specimens and/or imaging examinations. For prognosis, we retrospectively collected data from medical records in our institute.

### 2.3. Grouping of Patients with Pancreatic Cancer According to Their Diagnostic Approach 

We divided the patients with PC into eight groups according to how PC was diagnosed (Figure 1). These groups comprised patients who were diagnosed based on 1. symptoms from biliary obstruction (e.g., jaundice), 2. other symptoms (e.g., upper abdominal pain, back pain, appetite loss, weight loss, strong fatigue, and diarrhea), 3. new-onset or rapid worsening of diabetes mellitus, 4. high serum level of carbohydrate antigen 19-9 (CA19-9), 5. computed tomography, magnetic resonance imaging, or PET examination for other diseases, 6. surveillance of IPMN, 7. US during medical checkup of asymptomatic individuals (hereinafter referred to as US medical checkup), and 8. other reasons. In this analysis, we defined patients in group 1 or 2 as symptomatic group and 3–8 as asymptomatic group (Figure 1). Hereinafter, we also refer to patients in group 7 as the US medical checkup group. We show the 2 cases noticed by US medical checkup (Figure 2).

### 2.4. Evaluations

We analyzed the backgrounds, clinical stages, excision ratio, and prognosis of the participants and compared the differences between the symptomatic group and asymptomatic groups and between the symptomatic group and each of the other groups. 

### 2.5. Predictive Factors of Operable Pancreatic Cancers and Long-Term Prognosis in the Symptomatic Group and the Ultrasonography Medical Checkup Group

We collected the 259 patients with PC noticed by US medical checkup group (17 patients) and symptoms (242 patients). We analyzed the characteristics of resected cases among the 259 patients. In addition, we performed survival analysis for these 259 patients using Cox regression hazard model. The value of CA19-9, 425 (U/mL) and NLR, 3.5, was based on the median value of patients in groups 1, 2, and 7. The value of BMI, 17.5 (kg/mm^2^) was border between skinny and normal.

### 2.6. Details of Patients Diagnosed through Ultrasonography during Medical Checkup

We analyzed the details of patients diagnosed through US medical checkup. Place where PC was found, doctor’s specialty, patient’s comorbidities, ultrasonographic findings, clinical or pathological stage, therapy, and prognosis were described.

### 2.7. Statistical Analyses

Fisher’s exact test was used to compare categorical variables, and the Welch’s *t*-test and Median test were used to compare quantitative data where appropriate. Logistic regression analysis was performed to identify independent predictors of resectable PCs in groups 1,2,7. The log-rank test with the Kaplan–Meier method was used to evaluate survival in a univariate analysis, and a Cox regression hazard model was used for multivariate analysis to identify factors associated with prognosis. All statistical analyses of the recorded data were performed using the Excel statistical software package (Ekuseru-Toukei, version 2015; Social Survey Research Information Co., Ltd., Tokyo, Japan). *p* < 0.05 was considered as statistically significant.

## 3. Results

### 3.1. Patient Characteristics

Table 1 summarizes the clinical features of the 374 patients with PC (242 symptomatic patients and 132 asymptomatic patients; 192 men, 51.3% and 182 women, 48.7%) with a median age of 74 years (range, 34–105 years). The proportion of patients with any of the three comorbidities (hypertension, diabetes mellitus, and hyperlipidemia) was 70.9%. There were more patients with diabetes mellitus, hypertension, hyperlipidemia, history of malignancy, and history of smoking in the asymptomatic group than in the symptomatic group. The proportion of PC localized in the pancreatic tail in the 374 patients was 17.9%, and the excision ratio was 36.6%.

Patients in the asymptomatic group had significantly smaller tumor size (median tumor size: 28 mm vs. 39 mm, *p* < 0.001), earlier stage of PC (total proportion of stages 0, 1, and 2: 72.7% vs. 28.1%, *p* < 0.001), and higher excision ratio (62.1% vs. 23.1%, *p* < 0.001) than patients in the symptomatic group. In the asymptomatic group, CA19-9 and neutrophil to lymphocyte ratio (NLR) were lower and prognostic nutrition index (PNI) was higher than in symptomatic group (median CA19-9: 104 U/mL vs. 557 U/mL, *p* <0.01, NLR: 2.8 vs. 3.6, *p* < 0.001, and PNI: 48.8 vs. 46.5, *p* < 0.01, respectively).

### 3.2. Patient Characteristics in Each Group According to How Pancreatic Cancer Was Diagnosed

The characteristics of patients with PC according to the eight groups are described in Table 2. The other approaches used to diagnose PC in patents in group 8 are described in Appendix A. 

The proportion of patients with early stage PC (Stages 0, 1, and 2) and the excision ratio were significantly higher in each of groups 3–7 than in the symptomatic group. The excision ratio and proportions of stages 0, 1, and 2 were significantly higher in the US medical checkup group than in the symptomatic group (58.8% vs. 23.1%, *p* < 0.01 and 70.6% vs. 28.1%, *p* < 0.001, respectively). Further, NLR as a prognostic factor of PC was significantly better in the asymptomatic groups including the US medical checkup group compared with the symptomatic groups. 

### 3.3. Patients’ Prognosis in Each Group According to How PC Was Diagnosed

The median survival time (MST) in the symptomatic group was significantly shorter than that in the asymptomatic group (312 days vs. 919 days, *p* < 0.001; Figure 3). All groups in the asymptomatic group showed a significantly longer MST compared with that shown in the symptomatic group (Figure 4). Furthermore, MST and 5-SR in the US medical checkup group was better than that in the symptomatic group (1,764 days vs. 312 days, *p* < 0.001 and 42.2% vs. 9.4%, *p* < 0.001, respectively).

### 3.4. Excision Ratio and Prognosis in the Symptomatic Group Plus US Medical Checkup Group

The univariate analysis of patients in symptomatic group plus US medical checkup group showed more resected cases in the US medical checkup group (*p* < 0.01), females (*p* = 0.045), IPMN-derived carcinoma cases (*p* = 0.03), patients with normal or high body mass index (BMI) (BMI ≥ 18.5; *p* < 0.001), and patients with CA19-9 < 425 (*p* < 0.001). The multivariable analysis showed more resected cases in the US medical checkup group (*p* = 0.04), females (*p* < 0.01), patients with normal or high BMI (BMI ≥ 18.5; *p* = 0.02), and patients with CA19-9 < 425 (*p* < 0.001) (Table 3).

In the multivariate analysis using the Cox regression hazard model, there were significantly better prognoses in patients in the US medical checkup group (*p* < 0.01), with normal or high BMI (*p* < 0.01), with low CA19-9 (*p* < 0.001), and with low NLR (*p* < 0.01) (Table 4).

### 3.5. Details of Patients Found through Medical Checkup with Abdominal Ultrasonography

Details and summary of patients in the US medical checkup group are described in Appendix A, Table 2 and Table 5. There were three patients for which PC was identified at health screening centers and 12 PCs were identified during regular clinic visits. The 12 PCs were detected in clinics that had the machines and techniques to perform US and the specialties of all the clinicians were internal medicine, and the subspecialties in 10 of the 12 were gastroenterology. Thirteen out of 17 patients (76.5%) had any of the following basal diseases (hypertension, diabetes mellitus, and hyperlipidemia). The performance status of all the 17 PCs was 0 for all. Three of the 17 PCs were located in the pancreatic tail (17.6%) (Table 2) and all of the three pancreatic tumors were not detected using US. Two of the 17 were found as metastatic tumor of liver. The most frequent findings that indicated the presence of PC was dilation of the main pancreatic duct (MPD; 10/17; Appendix A and Table 5). Median tumor size was small in US medical checkup group compared with the symptomatic group (median: 20 mm [0–52] vs. 34 mm [0–128]), but there was no statistically significant difference (*p* = 0.08) (Table 2). The proportion of patients in stages 0, 1, and 2 and the excision ratio were significantly higher in US medical checkup group than in the symptomatic group (stages 0, 1, and 2: 70.6% vs. 28.1%, *p* < 0.001 and excision ratio: 58.8% vs. 23.1%, *p* < 0.001; Table 2). In the resected cases, two were at stage 0; one was identified by MPD dilation induced by stenosis associated with pancreatic intraepithelial neoplasia-3 and the other by MPD dilation induced by MPD-IPMN (IPMN with high-grade dysplasia; Appendix A and Table 5). Further, there were 8 resected cases at stage 2 (2a, 4 and 2b, 4) and 7 out of the 8 cases were alive with no relapse (539–2690 days). In addition, there were 3 patients with no relapse over 7 years (Appendix A and Table 5).

## 4. Discussion

In this study, we showed that PCs identified by US during medical checkup for asymptomatic individuals had better excision ratio and more excellent prognosis than that observed in PCs identified through symptoms. These results suggest that screening for PC using US in asymptomatic individuals might be one of the effective modalities for improved prognosis with the medical treatment of PC.

Patients with PC have an extremely poor prognosis. Further, PC remains the fourth leading cause of cancer-related deaths in Japan and the United States, with increasing incidence rates [9,10]. Thus, overcoming of PC is an urgent matter.

It has been increasingly recognized that the prognosis of patients with early stage PC is favorable [11,12], and PCs that can radically cured are stage 0 (in situ) and stage IA in Union for International Cancer Control 8th edition. The 5-SRs of stage 0 and stage IA are 85.8% and 68.7%, respectively [12,13]. However, the corresponding proportion of stages 0 and 1A cases accounts for only 1.7% and 4.1%, respectively [12,13]. Especially, PC at stage 0 does not form mass, and the carcinoma cannot be identified on imaging modalities, so stage 0 is now diagnosed using pancreatic juice cytology [14,15], focusing on indirect findings such as MPD dilatation and/or stenosis, cyst formation, focal fat deposition, and focal atrophy of the pancreas [13,16,17,18]. Many researchers have been making effort to identify early stage PC. Finding of PC patients with stage 0 or IA are increasing, but their proportion in all PCs is currently still low. This may explain that any imaging examinations (e.g., CE-CT, MRCP, EUS, and US) are needed as an indicator for attempting pancreatic juice cytology.

In 2012, the 5-SR of resected pancreatic cancers in Japan was approximately 20% [1]. However, the progress of adjuvant chemotherapy [19], neoadjuvant chemotherapy [20], excision technique, perioperative management [21,22], and chemotherapy [23,24] at the time of relapse is obviously improving the prognosis of patients with operable PC. Adjuvant chemotherapy for resected PCs using Tegafur Gimeracil Potassium (S-1) showed a 5-SR of 44.1% [19] and neoadjuvant chemotherapy using gemcitabine and S-1 for resectable PCs also showed a 2-year overall survival rate of 63.7% [20]. Further, disease specific 5-SR and recurrence free 5-SR were 52% and 40%, respectively, in resected PCs at our institute (N = 98, 2015–2021, unpublished data). Thus, identifying operable PCs might induce significant hope of better prognosis.

Typically, the effective way to detect cancers earlier might be cancer screening for asymptomatic individuals. Cancer screening is recommended by the Ministry of Health and Welfare of Japan for five cancers including lung, stomach, breast, colon, and uterus neck cancers, but is not recommended for pancreatic cancer. The recommendation for cancer screening may not only be due to the improvement of mortality but also the avoidance of unnecessary examinations and therapies. To identify PCs early, routine CE-CT, MRCP or EUS examination should be performed more than twice a year. However, US performed once a year might be a better modality for public pancreatic cancer screening, considering its non-invasiveness, ease of use, and lower cost. One additional advantage of US is the ability to find other abdominal cancers including liver, kidney, biliary tract organs, and urinary tract organs. The total numbers of these cancers are more than the total numbers of esophagus and stomach cancers in Japan [25]. In this study, asymptomatic patients whose PC was identified through US during medical checkup had better prognosis than patients noticed by symptoms. Thus, US for asymptomatic individuals might be recommended as one of the pancreatic cancer screening tools to improve the prognosis of PC.

Currently, PCs are identified earlier [3] based on new-onset diabetes mellitus [26], surveillance for IPMN [4,27], and FH of PC [28,29]. Our study showed that patients with PC identified based on new-onset or worsening of diabetes mellitus had better excision ratio and prognosis than patients identified based on symptoms of PC, making this algorism very important. The proportion of patients with PC identified by worsening of diabetes mellitus is reported as 4–5% [30], and our result (22/374 = 5.9%) is similar to this report. The use of new-onset or worsening of diabetes mellitus as an indicator for detecting PCs should be carefully monitored. Most investigators screen patients with IPMN using MRCP, EUS, and CE-CT once or twice a year [4,27,31,32,33,34,35]. If this rigorous surveillance is performed, it is natural that the patients are identified in an earlier stage and have better prognosis than that observed in patients identified at the symptomatic stage. Even in this study, the best prognosis was obtained in the group with IPMN who were screened for PC, and this result is thought to be natural because they had regularly CE-CT and MRCP examination twice a year.

Thinking from another point of view, the proportion of PDACs from the surveillance of patients with IPMN in this study was very low (12/374, 3.5%). Further, the lifetime carcinogenic rate of PC was reported at frequency of 2.6% in males and 2.5% in females [5]. If the carcinogenic rate of IPMN is 2.0–10% [4], it implies low carcinogenesis due to IPMN. The ratio of patients with a FH of 1st degree PC was 9% in this study and 3–8.7% in previous reports [5,6,7]. These facts indicate that most PCs could not detected only through surveillance of patients with IPMN and/or FH. We might need to identify PCs from individuals with none of the risks or with a small risk of PCs.

US is effective for outpatient care because of its convenience and non-invasiveness. Its sensitivity and specificity ranges for detecting PCs are broad (48–89% and 40–91%, respectively), and there are some differences according to operators, participants, and machines [36,37,38,39]. However, there were positive reports that its sensitivity for PCs under 10 mm was 50% and over 30 mm was 95.8% [36,37,38,39]. In addition, recent reports have shown that MPD dilatation as an indirect abnormality of PCs were identified by US in 62–75% of patients in PC stage 0 [18,40], 61% in stage 1A [40], and 74.3% in stage 1 [18]. MPD dilatation is caused by stasis of pancreatic juice in the downstream side of MPD not only due to invasive carcinoma but also due to carcinoma in situ [18,40]. Moreover, most PCs occur in the pancreatic head (78%) [41]. Thus, there is a chance of detecting tumor or MPD dilatation by US. In our analysis, there were two patients with PC stage 0 identified by MPD dilatation on ultrasonographic findings. One had MPD stenosis in the pancreatic head due to pancreatic intraepithelial neoplasm-3. The other one PC was identified by MPD dilatation (10 mm) in the pancreatic body and was diagnosed based on MPD-IPMN with high-grade dysplasia after surgery. Therefore, US might be useful for identifying early stage (stage 0) PCs. In addition, 11 patients in US medical checkup group had undergone resection, and 8 patients were in stage 2. Of the 8 patients, 7 were alive with no relapse and with a survival time ranging from 539 days to 2690 days at the time of this analysis. In addition, 3 of 7 were alive at over 7 years. This shows some PCs at stage 2 have the potential of being radically cured, which could be explained by progress in chemotherapy and surgical skills.

The disadvantage of US is that its use in some location, especially in the pancreatic tail makes the findings difficult to describe. In our study, three tumors were located in the pancreatic tail, and all could not be described by US. Hepatic metastasis was detected in two of the 3 patients by US. Ashida et al. [42] reported how pancreatic tail tumors can be described using repletion of the stomach by drinking tea with milk and obtained good result. Efforts need to be channeled towards describing tumors in the pancreatic tail more clearly using regular observations of the pancreas from the left lateral region, the repletion of stomach using fluids, and others. Another disadvantage of US is that the number of clinics that provide this service is low, and most clinic doctors do not have the technique for screening abdominal organs on US. In this study, most PCs were identified by clinicians whose subspecialty was gastroenterology. In addition, we are very sorry that US aimed at medical checkup is not covered by insurance under the medical insurance system of Japan.

In Onomichi city in Hiroshima prefecture and Yamanashi prefecture, clinics and medical examination centers work closely with referral centers, and the PC discovery rate has been increasing, and some patients with stage 0 and 1 are identified [13,15,43,44]. In Onomichi city, clinic doctors are performing US for patients with multiple risk factors of PC (FH, diabetes mellitus, smoking, heavy drinking, obesity, et.al) and refer to referral centers if there are abnormal findings (mass, cyst, or MPD dilatation). Naturally, public cancer screening for upper abdominal organs (liver, biliary tract, kidney, pancreas, and spleen) by US is performed in Onomichi city. In our analysis, there were many PC patients with any disease such as hypertension, diabetes mellitus, and hyperlipidemia (265/374, 70.9%). Thus, there might be big chance to detect asymptomatic PCs if US in clinics can be performed for them using medical insurance system or public cancer screening system.

Incident ratio of pancreatic cancer is rapidly increasing in individuals of 60’s and the incident ratio increase with age. In comparison with 50’s, the numbers of PCs are 3.3fold in 60’s, 4.5fold in 70’s, and 3.9fold in 80’s [45]. Thus, pancreatic cancer screening focusing on 60’s and 70’s might be useful for detecting PCs.

There are several limitations in this study. First, this is a retrospective analysis in a single center, so there are small number of cases in the analysis, and there may have been single center bias. Particularly, majority of the patients with PC in our analysis were older than that observed in other high-volume centers because of the increasing aging population in Kure city. Second, we could not show the detectability of pancreatic cancer with US because most patients were diagnosed with CT and underwent EUS, not US, as additional examinations. Thirdly, there might have been a bias in the grouping of patients into the 8 groups because the groupings were based on reports from referral letters. Finally, we included patients who were diagnosed in past years, so there were some differences in the diagnostic modalities and standard therapy, which were based on the year in which patients were diagnoses and how these factors influenced prognosis.

In 2019, The United States Preventive Service Task Force reported [8] that there was no evidence that screening for PC improves disease-specific morbidity or mortality, and they provided no recommendation for PC screening in asymptomatic adults, considering the low incidence ratio of PC in the general population, the uncertain accuracy of current candidate screening tests (CT, MRCP, and EUS), and poor prognosis of PC even when it is treated at an early stage. In contrast, some reports have shown the cost effectiveness of US in identifying PCs [46,47]. Further, Tanaka reported better excision rate (76.9%) in patients for which PC was identified during medical checkup [48]. We suggest that pancreatic cancer screening by any modalities such as US, focusing on the older population (60–75 years old) might be more efficient.

## 5. Conclusions

PCs identified by US during medical checkup for asymptomatic individuals had more excellent prognosis than that observed in PCs identified through symptoms. The public should be educated about the importance of PC screening in asymptomatic individuals, and we need to accumulate the evideces that show efficacy of cancer screening for PC using any modalities such as CE-CT, MRCP, EUS, and US.

## Figures and Tables

**Figure 1 diagnostics-12-02913-f001:**
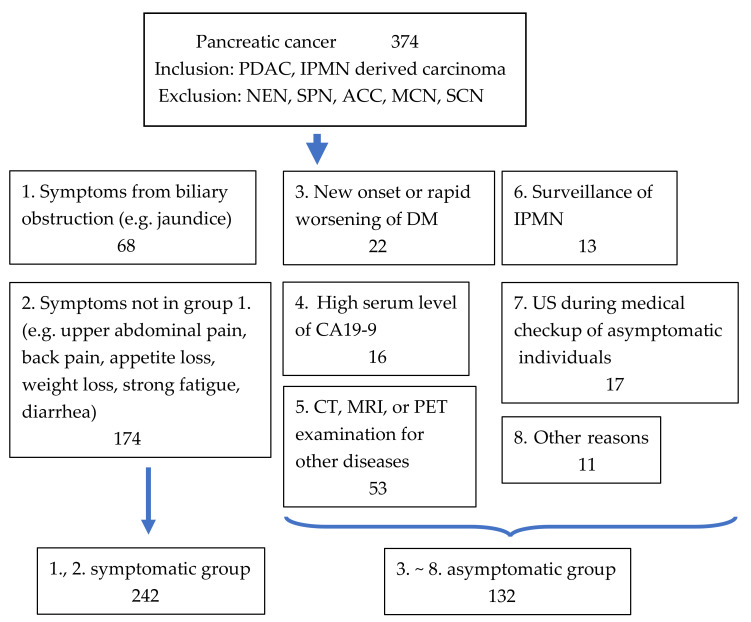
Patient flow diagram. We collected 374 patients with pancreatic cancers (PCs) including those with pancreatic ductal adenocarcinoma (PDAC) and intaraductal papillary neoplasm (IPMN) with high-grade dysplasia, and IPMN associated with invasive carcinoma and excluding those with neuroendocrine neoplasm (NEN), solid pseudo-papillary neoplasm (SPN), acinar cell carcinoma (ACC), mucinous cystic neoplasm (MCN), and serous cystic neoplasm (SCN). We divided patients with PC into 8 groups according to how PC was diagnosed. Patients in group 1 or 2 were defined as the symptomatic group and 3–8 as the asymptomatic group. DM:diabetesmellitus, CA19-9: carbohydrate antigen 19-9, CT: computed tomography, MRI: magnetic resonace imaging, PET: positron emission tomography, IPMN: intraductal papillary mucinous neoplasm, US: abdominal ultrasonography.

**Figure 2 diagnostics-12-02913-f002:**
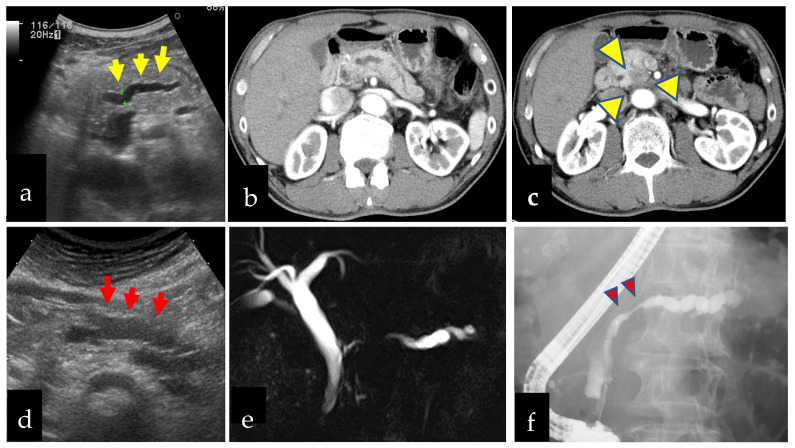
Two cases of pancreatic cancer found by medical checkup by abdominal ultrasonography. (**a**–**c**): Sixty-five-year-old man had a dilatation of the main pancreatic duct (MPD) in the body (yellow arrows) and underwent enhanced computed tomography and pancreatic head cancer was found (yellow arrowheads). He had surgery and a diagnosis of stage 2b (UICC 8th) and was alive without relapse 2621 days after surgery. (**d**–**f**): An eighty-year-old women had a dilatation of MPD in the pancreatic body (red arrows) and underwent magnetic resonance cholangiography and focal poor rendering of MPD was detected. Endoscopic retrograde pancreatography showed focal stenosis of MPD in the pancreatic body (red arrowheads). Serial pancreatic juice cytology showed atypical cells suspect of adenocarcinoma and was diagnosed with pancreatic intra-epithelial neoplasm-3 with surgery.

**Figure 3 diagnostics-12-02913-f003:**
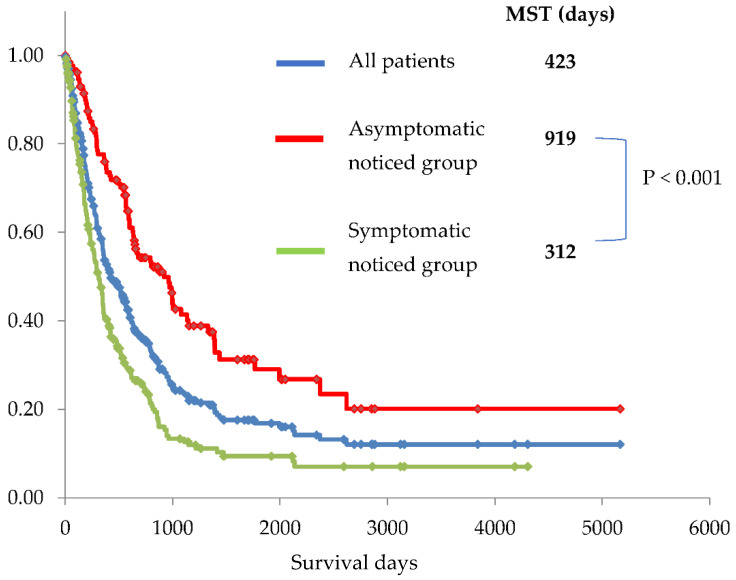
Kaplan–Meier curves for all patients (blue line), symptomatic group (group 1 + 2) (green line), and asymptomatic group (group 3–8) (red line). The median survival time (MST) was significantly longer, and 5-year overall survival rate was significantly higher in the asymptomatic group than in the symptomatic group (312 days vs. 919 days and 5.4% vs. 29.0%, *p* < 0.001, respectively).

**Figure 4 diagnostics-12-02913-f004:**
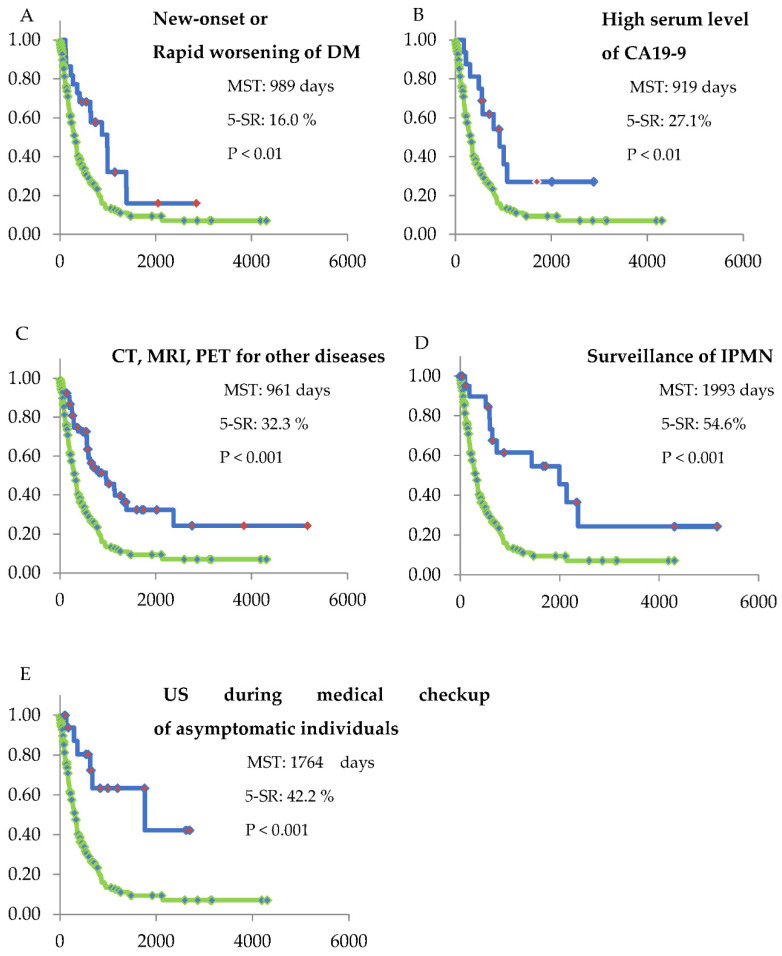
Kaplan–Meier curves of the symptomatic group and each of the asymptomatic groups. The green line shows the Kaplan–Meier curve of the symptomatic group. Statistical analysis for survival was performed in each group compared with the symptomatic group. The median survival time (MST), 5-year survival rate (5-SR), and the *p*-value are shown in each Figures ((**A**): group 3, (**B**): group 4, (**C**): group 5, (**D**): group 6, and (**E**): group 7). All of the asymptomatic groups showed a significantly longer MST compared with that shown in the symptomatic group. The horizontal axis shows survival days.

**Table 1 diagnostics-12-02913-t001:** Patients’ characteristics in all patients.

	All Patients	Symptomatic Group	Asymptomatic Group	*p*-Value #
Patients’ number	374	242	132	
Male, *n* (%)	192 (51.3)	126 (52.0)	66 (50)	0.75
Age, median (range), years	74 (34–105)	72 (34–105)	76 (44–98)	<0.01
Comorbidities				
Diabetes mellitus, *n* (%)	120 (32.1)	56 (23.1)	64 (48.5)	<0.001
Hypertension, *n* (%)	162 (43.3)	93 (38.4)	69 (52.3)	<0.05
Hyperlipidemia, *n* (%)	76 (20.3)	37 (15.3)	39 (29.5)	<0.01
Any of the above 3 diseases, *n* (%)	265 (70.9)	160 (66.1)	105 (79.5)	<0.01
History of other cancer, *n* (%)	70 (17.7)	34 (14.0)	36 (26.9)	<0.01
History of heavy drinking (ethanol ≥100 g/day)	19 (5.1)	9 (3.7)	10 (7.6)	0.13
History of smoking, *n*/N (%)	197/373 (52.8)	115/241 (47.7)	82/132 (62.1)	<0.01
Family history of PC				
(≤1st degree), *n*/N (%)	27/289 (9.3)	15/180 (8.3)	12/109 (11.0)	0.53
(≤2nd degree), *n*/N (%)	30/289 (10.4)	18/180 (10)	12/109 (11.0)	0.84
PDAC, IPMN-derived carcinoma, *n*	355, 19	235, 7	120, 12	<0.05
Localization of PC				
uncus, head, groove, head~body, body, body~tail, tail	35, 119, 8, 11, 104, 30, 67	24, 81, 9, 6, 64, 19, 39	7, 38, 3, 5, 40, 11, 28	
tail, *n* (%)	67 (17.9)	39 (16.1)	28 (21.2)	0.26
Tumor size * median (range), mm	34 (0–128)	66 (0–128)	25 (0–100)	<0.001
Clinical or pathological Stage (UICC 8th)				
0, 1, 2, 3, 4	12, 8, 144, 41, 169	1, 1, 66, 35, 139	11, 7, 78, 6, 30	
0, 1, 2, *n* (%)	164 (43.9)	68 (28.1)	96 (72.7)	<0.001
Therapy				
BST, *n* (%)	76 (20.3)	57 (23.6)	19 (14.4)	<0.05
Chemotherapy	158	128	30	
Radiation	2	1	1	
Excision, *n* (%)	138 (36.9)	56 (23.1)	82 (62.1)	<0.001
BMI, median (range), kg/mm^2^	21.9 (13.6–35.2) *n* = 371	21.6 (14.3–35.2) *n* = 240	22.5 (13.6–34.3) *n* = 131	0.13
BMI < 18.5, *n* (%)	67 (18.1)	48 (20)	19 (8.2)	0.06
18.5 ≤ BMI < 25	323	155	77	
25 ≤ BMI	72	37	35	
CA19-9, median (range), U/mL	239 (<2–26,165,454)	557 (<2–26,165,454)	104 (<2–7,575,434)	<0.01
AMY, median (range), U/L	68 (12–902) *n* = 373	63 (12–902) *n* = 242	77 (13–372) *n* = 131	<0.05
Alb, median (range), g/dL	4.0 (12–90.2) *n* = 371	4.0 (2.2–5) *n* = 241	4.1 (2.5–5.2) *n* = 130	<0.01
NLR, median (range)	3.4 (0.6–27.7) *n* = 372	3.6 (0.6–27.7) *n* = 241	2.8 (0.69–12.5) *n* = 131	<0.001
PNI, median (range)	47.6 (26.6–80.1) *n* = 370	46.5 (26.6–61.7) *n* = 240	48.8 (28.3–80.1) *n* = 130	<0.01

* Tumor size was calculated using the solid part. We had several data defectiveness, and “*n*” in the table shows analyzed patient number and “*n*/N” in the table shows positive number/analyzed number. **#** Statistical analysis was performed to compare the differences between the asymptomatic and symptomatic groups. PC: pancreatic cancer, PDAC: pancreatic ductal adenocarcinoma, IPMN: intraductal papillary mucinous neoplasm, UICC 8th: Union for International Cancer Control 8th edition, BST: best supportive therapy, BMI: body mass index, CA19-9: serum level of carbohydrate antigen 19-9, AMY: serum level of amylase, Alb: serum level of albumin, NLR: neutrophil to lymphocyte ratio, PNI: prognostic nutrition index.

**Table 2 diagnostics-12-02913-t002:** Patient’s characteristics in each group.

Group	1	2	1, 2	3	4	5	6	7
Patients’ number	68	174	242	22	16	53	13	17
Male, *n* (%)	29 (42.6)	97 (55.7)	126 (52.1)	9 (40.9)	4 (25) *	32 (60.4)	10 (76.9)	8 (47.1)
Age, median (range), years	76 (41–105)	71 (34–93)	71 (34–105)	74 (44–87)	83 (66–89) *	76 (45–89)	81 (71–86) *	75 (59–86)
IPMN-derived carcinoma, (vs. PDAC), *n* (%)	1 (1.5)	6 (3.4)	7 (2.9)	0 (0)	1 (1.9)	6 (11.3)	3 (23.1) *	1 (5.9)
Localization of PC								
uncus, head, groove, head~body, body, body~tail, tail	3, 49, 7, 3, 5, 1, 0	21, 32, 2, 3, 59, 18, 39	24, 81, 9, 6, 42, 19, 39	0, 8, 0, 2, 5, 2, 5	2, 4, 0, 0, 5, 1, 4	3, 18, 0, 1, 17, 6, 8	0, 4, 0, 0, 4, 0, 5	2, 2, 1, 2, 7, 0, 3
tail, *n* (%)	0 (0)	39 (22.4)	39 (16.1)	5 (22.7)	4 (25)	8 (15.1)	5 (38.5)	3 (17.6)
Tumor size **, median (range), mm	30 (0–63)	36 (0–128)	34 (0–128)	25 (0–77) #	28 (18–55) *	26 (0–100) ●	17 (0–40) #	20 (0–52)
Clinical or pathological stage (UICC 8th)								
0, 1, 2, 3, 4	0, 0, 37, 7, 24	1, 1, 29, 28, 115	1, 1, 66, 35, 139	1, 1, 15, 1, 4	0, 0, 13, 2, 1	4, 5, 27, 3, 14	3, 0, 8, 0, 2	2, 1, 9, 0, 5
0, 1, 2, *n* (%)	37 (54.4)	31 (17.8)	68 (28.1)	17 (77.3) ●	13 (81.3) ●	36 (67.9) ●	11 (84.6) ●	12 (70.6) ●
Therapy								
BST, *n* (%)	28 (41.2)	29 (16.7)	57 (23.6)	1 (4.5)	2 (12.5)	12 (22.6)	2 (15.4)	0 (0) *
Chemotherapy	14	114	128	6	2	12	0	7
Radiation	0	1	1	0	0	1	0	0
Excision, *n* (%)	26 (38.2)	30 (17.2)	56 (23.1)	15 (68.2) ●	12 (75) ●	28 (52.8) ●	11 (84.6) ●	10 (58.8) ●
BMI, median (range), kg/mm^2^	21.8 (14.4–32.7)	22.1 (14.3–35.2) *n* = 173	21.6 (14.3–35.2) *n* = 241	22.1 (15.3–28.8)	24.1 (16.3–34.3)	21.2 (13.6–31.9)	21.7 (16.2–25.6)	21.6 (18.0–29.8)
BMI < 18.5, *n* (%)	15 (22.1)	15 (8.6)	30 (12.4)	5 (22.7)	1 (6.3)	11 (20.8)	1 (7.7)	1 (5.9)
CA19-9, median (range), U/mL	231 (<2–194,660) *n* = 67	824 (<2–26,165,454) *n* = 172	557 (<2–26,165,454) *n* = 239	312 (<2–10,334)	391 (43–3618)	28 (<2–7,574,431)	15 (<2–25,550) ●	9 (<2–21,945) ●
AMY, median (range), U/mL	72 (15–517)	61 (12–902)	63 (12–902)	68 (26–267) *n* = 21	86 (33–124)	88 (27–372) *	77 (20–192)	60 (27–286)
Alb, median (range), g/dL	3.7 (2.2–4.9)	3.9 (2.2–5.0) *n* = 173	4.0 (2.2–5.0) *n* = 241	4.2 (3.2–5.1) *n* = 21	4.2 (3.5–4.5)	4.0 (3.2–5.2)	4.2 (3.2–4.4) *n* = 12	4.0 (2.9–4.8)
NLR, median (range)	3.9 (0.9–27.7)	3.6 (0.6–22.6) *n* = 173	3.6 (0. 6–27.7) *n* = 241	2.5 (0.9–11)	2.9 (1.2–4.0) *	2.8 (0.7–12.5) #	1.9 (0.7–3.6) *n* = 12 #	2 (0.9–5.3) *
PNI, median (range)	44.7 (27.6–57.2) *n* = 66	47.7 (7.7–61.7) *n* = 172	46.5 (7.7–61.7) *n* = 238	49.1 (38.9–62.2) *n* = 21 *	49.9 (39.9–58)	48.1 (28.3–80.1)	50.2 (45.9–59.5) *n* = 12 #	47.2 (39–60.5)

Each group comprised patients identified through 1. symptoms of biliary obstruction, 2. symptoms that were not in group 1, 3. new-onset or rapid worsening of diabetes mellitus, 4. high serum carbohydrate antigen 19-9 (CA19-9) level, 5. computed tomography, magnetic resonance imaging, or positron emission tomography examination for other diseases, 6. surveillance of IPMN, and 7. US during medical checkup of asymptomatic individual. ****** Tumor size was calculated using solid part. We compared the differences of items between the asymptomatic noticed group (1 and 2) and each group (3–7) (●: *p* < 0.001, #: *p* < 0.01, *****: *p <* 0.05). We had several data defectiveness, and “*n*” in the table shows analyzed patient number and “*n*/N” in the table shows positive number/analyzed number. IPMN: intraductal papillary mucinous neoplasm, PC: pancreatic cancer, UICC 8th: Union for International Cancer Control 8th edition, BST: best supportive therapy, BMI: body mass index, AMY: serum level of amylase, Alb: serum level of albumin, NLR: neutrophil to lymphocyte ratio, PNI: prognostic nutrition index.

**Table 3 diagnostics-12-02913-t003:** Comparison of characteristics between resected and unresected patients in Groups 1, 2, and 7.

	Unresected	Resected	Univariate Analysis (*p*-Value)	Multivariate Analysis	
				*p*-Value	Odds Ratio (95%CI)
Patients’ number	193	66			
Female, *n* (%)	94 (48.7)	40 (60.6)	0.045	<0.01	2.36 (1.26–4.45)
Age, median (range), years	73 (34–105)	72 (42–86)	0.78		
Age, ≥75 years old, *n* (%)	87 (45.1)	25 (37.9)	0.32	0.20	0.66 (0.34–1.26)
Group 7 (vs. Group1,2), *n* (%)	7 (3.6)	10 (15.2)	<0.01	0.04	3.31 (1.08–10.11)
Diabetes mellitus, *n* (%)	43 (22.3)	16 (9.1)	0.74		
Any of the 3 diseases (diabetes mellitus, hypertension, hyperlipidemia), *n* (%)	105 (54.4)	40 (24.2)	0.09		
History of other cancer, *n* (%)	26 (13.5)	11 (16.7)	0.39		
History of heavy drinking (ethanol ≥100 g/day)	7 (3.6)	3 (4.5)	0.72		
History of smoking, *n*/N (%)	96/192 (50)	27 (40.9)	0.25		
Family history of PC (≤1st degree), *n*/N (%)	13/145 (9.0)	3/50 (6)	0.77		
IPMN-derived carcinoma, PDAC, *n*	3, 190	5, 61	0.03	0.10	3.81 (0.78–18.67)
Localization of PC tail, *n* (%)	36 (18.7)	6 (9.1)	0.08		
BMI, median (range), kg/mm^2^	21.1 (14.3–35.2) *n* = 191	23.0 (15.0–32.7)	<0.001		
BMI (kg/mm^2^) ≥ 18.5, *n*/N (%)	146/191 (76.4)	62/66 (93.9)	<0.01	0.02	3.76 (1.24–11.43)
CA19-9, median (range), U/mL	1352 (1–215,454) *n* = 190	165 (1–4941)	<0.001		
CA19-9 (U/mL) ≥ 425	109/190 (5 7.4)	17/66 (25.8)	< 0.001	<0.001	0.31 (0.16–0.59)
AMY (U/L), median (range),	62.0 (12–902)	68.5 (24–664)	0.15		

Groups 1 and 2 included patients with pancreatic cancer (PC) identified through symptoms of biliary obstruction and symptoms that were not in group 1, respectively. Group 7 included patients with PC identified through abdominal ultrasonography (US) during medical checkup of asymptomatic individuals. The value of CA19-9, 425 (U/mL) was based on the median values of patients in Groups 1, 2, and 7. We had several data defectiveness, and “*n*” in the table shows the analyzed patient number and “*n*/N“ show positive number/analyzed number. IPMN: intraductal papillary mucinous neoplasm, PDAC: pancreatic ductal adenocarcinoma, BMI: body mass index, CA19-9: serum level of carbohydrate antigen 19-9, AMY: serum level of amylase, CI: confidence intervals.

**Table 4 diagnostics-12-02913-t004:** Prognostic factors in patients in groups 1, 2, 7.

	Multivariate Analysis	
	*p*-Value	Hazard Ratio (95%CI)
Female sex	0.16	0.81 (0.60–1.09)
Age, ≥75 years old	0.10	1.29 (0.98–1.75)
Group7 (vs. group 1or 2)	<0.01	0.36 (0.17–0.78)
IPMN-derived carcinoma (vs. PDAC)	0.13	0.49 (0.19–1.22)
BMI (kg/mm^2^) ≥18.5	<0.01	0.60 (0.42–0.87)
CA19-9 (U/mL) ≥425	<0.001	1.69 (1.24–2.30)
NLR ≥ 3.6	<0.01	1.58 (1.18–2.11)

Group 1 and 2 included patients with pancreatic cancer (PC) identified through symptoms of biliary obstruction symptoms and symptoms that were not in group 1, respectively. Group 7 included patients with PC identified through abdominal ultrasonography during medical checkup of asymptomatic individuals. Statistical analysis was performed using Cox regression hazard model. The value of CA19-9, 425 (U/mL) and NLR, 3.5, was based on the median value of patients in groups 1, 2, and 7. IPMN: intraductal papillary mucinous neoplasm, PDAC: pancreatic ductal adenocarcinoma, BMI: body mass index, CA19-9: serum level of carbohydrate antigen 19-9, NLR: neutrophil to lymphocyte ratio, CI: confidence intervals.

**Table 5 diagnostics-12-02913-t005:** Summary of patients in group 7.

Patients’ Number	17
Place where PC was found	
Clinic going regularly	12
Health screening center	3
Referral center (our hospital)	2
Specialty of doctors in clinic	
Internal medicine, *n*/N	12/12
Subspecialty	
Gastroenterology	10
Respiratory medicine	1
Unknown	1
Comorbidities	
Diabetes Mellitus	3
Hypertension	9
Hyperlipidemia	10
Each of above 3 diseases, *n* (%)	13 (76.5)
Performance status	
0, 1, 2, 3, 4	17, 0, 0, 0, 0
Findings of ultrasonography	
Tumor in pancreas	6
Main pancreatic duct dilatation	10
Cyst in pancreas	1
Tumor in liver	2
Patients with operation	11
Pathological stage, 0, 1, 2a, 2b, 3, 4 (UICC 8th)	2, 1, 4, 4, 0, 0
No relapse in stage 2, *n*/N (%)	7/8 (87.5)
Days after surgery in 7 patients with no relapse in stage 2	539, 642, 834, 992, 2619, 2621, 2690

Group 7 included patients with PC identified through abdominal ultrasonography during medical checkup of asymptomatic individuals. UICC 8th: Union for International Cancer Control 8th edition.

## Data Availability

The datasets used and/or analyzed during the current study are available from the corresponding author on reasonable request.

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
