# Peer review of "Effectiveness of Abdominal Ultrasonography for Improving the Prognosis of Pancreatic Cancer during Medical Checkup: A Single Center Retrospective Analysis"

_diagnostics, 2022, doi:10.3390/diagnostics12122913_

Round 1
Reviewer 1 Report
I generally very much liked and appreciated the manuscript.
I have some minor concerns:
1. Adding a sonographic image of an incidentally detected pancreatic cancer and/or main duct dilatation, preferably together, can increase the quality.
2. In fig.1, both groups were named as symptomatic, please correct this.
Author Response
Answer to Reviewer 1
I really appreciate what you’ve done for my manuscript.
I’m very happy to get your kind comments.
I revised my manuscript along with your advices.
- Adding a sonographic image of an incidentally detected pancreatic cancer and/or main duct dilatation, preferably together, can increase the quality.
à Thank you for your advice. I added cases of PDAC found by abdominal ultrasonography in Figure 2.
- In fig.1, both groups were named as symptomatic, please correct this.
à Thank you for your advice. We corrected.
Reviewer 2 Report
Dear author
Thank you for the submission of your article to our journal. I’ve just read your article and found Manu problems as follows;
Major point
It is a well-known fact that asymptomatically discovered cancers, not limited to pancreatic cancer, have more early stage cases than symptomatic cancers and have a better prognosis. If you would like to stress the usefulness of US for detecting the pancreatic cancer, you should at least clarify the detectability of pancreatic cancer with US in your study.
Minor points
Your English needs extensive revision by some senior authors. I listed part of the strange points below.
Page 1-2
The expression “there is a need identify asymptomatic patients with PC” is strange.
Page 2
The sentence “Recently, to ensure early detection of PCs, attention has been paid to patients with new-onset or rapid worsening of diabetes mellitus and surveillance for individuals with intraductal papillary mucinous neoplasm (IPMN) and family history (FH) of PC [3]. “ is strange.
The sentence “These unfortunate results might be due to the fact that most patients with PC do not come from screening for IPMN and FH of PC, although these two factors are important indicators of PCs [4–7]. “ is strange.
The sentence “One reason is the report that there is no evidence of cancer screening improving the disease-specific morbidity or mortality of PC [8]. “ is strange.
The sentence “In this study, we analyzed the prognosis of patients with PC who were divided into eight groups according to how PC was diag- nosed and evaluated the usefulness of abdominal ultrasonography (US) during medi- cal checkups of asymptomatic individuals. “ is strange.
The sentence "For diagnosis, we first used surgical specimens, and imaging studies were used if surgery was not per- formed. ” is strange.
P3
Groups 1 or 2→ group 1 or 2
The terms “symptomatic noticed group” and “asymptomatic noticed group “ are strange.
P4
The expression “To further confirm the efficacy of US medical checkup compared to symptom identification” is strange.
The expression “we performed multivariate analysis of the excision ratio and prognosis of patients with were diagnosed by symptoms (242 patients) and by US medical checkup (17 patients), totaling 259 patients.” is grammatically strange.
The word “prescribed” is inappropriate.
P5
The expression “compared with that had by the symptomatic noticed group “ is strange.
The expression “CA19-9, neutrophil to lymphocyte ratio (NLR), prognostic nutrition index (PNI) possible prognostic factors of PC were better” is strange.
P7
The expression “In addition, the US medical checkup group had significantly lower proportion of patients with best supportive therapy than that had by the symptomatic noticed group.”
This sentence is just a paraphrasing of the previous sentence.
P9
The univariate analysis of patients in asymptomatic noticed group plus US medical checkup group showed more resected cases in the US medical checkup group (P<0.01).
What would you like to emphasize from this sentence?
CA19-9 <425
Why did you set the cutoff value of CA19-9 at 425?
P10
prognosis →prognoses
P11
Why did you set the cutoff value of BMI at 18.5.
The expression “”There were only three patients for which PC was identified at health screening centers” is strange.
P12
You used the sentence ”PC has the worst prognosis among all cancers, and its 5-SR is approximately 7.1% and 10% in Japan and the United States, respectively.” both in the introduction and discussion.
The expression “Thus, overcoming the burden of PC is an urgent issue.” is strange.
The expression “PCs that can radically cured are in Union for International Cancer Control stage 0 (in situ) and stage IA with 5-SRs of 85.8% and 68.7%, respectively [12,13]. “ is strange.
The expression “Patients with stage 0 or 1A are increasing” is strange.
P13
The expression “Adju- vant chemotherapy using Tegafur Gimeracil Potassium (S-1) showed a 5-SR of 44.1% [19] and neoadjuvant chemotherapy using gemcitabine and S-1 for resected PCs also showed a 2-year overall survival rate of 63.7% [20]. “ is strange.
The expression “as it is not required by the Ministry of Health and Welfare, Japan” is strange.
The expression “downregulation of mortality” is strange.
What’s the meaning of simpleness of US?
The expression “prolong the prognosis” is strange.
P14
What’s the meaning of "following fibrosis”?
The expression "Surprisingly, 3 were alive patients and without relapse for over 7 years and had the potential of achieving complete remission. ” is strange.
The expression “progression of chemotherapy and surgeons’ technique “ is strange.
Author Response
To Reviewer 2
I really appreciate what you’ve done for my manuscript.
I’m very happy to get your useful comments and advices.
I revised my manuscript along with your advices.
Major point
It is a well-known fact that asymptomatically discovered cancers, not limited to pancreatic cancer, have more early stage cases than symptomatic cancers and have a better prognosis. If you would like to stress the usefulness of US for detecting the pancreatic cancer, you should at least clarify the detectability of pancreatic cancer with US in your study.
---->
Thank you for your suggestion. You are right, but we could not show the detectability of pancreatic cancer with US in our study. Unfortunately, most patients were diagnosed with computed tomography and most patients had endoscopic ultrasonography and not US as additional examinations. Thus, we could not show the detectability of pancreatic cancer with US in this study.
We referred to this point as a limitation of our study.
As you say, the sensitivity and specificity of US for diagnosis of pancreatic cancer must be lower than EGD for a diagnosis of gastric cancer. Probably, this is one of the reasons that cancer screening for PC is not currently recommended in Japan and the United States. Thus, public cancer screening for upper abdominal organs including the pancreas is performed in few municipalities. Certainly, the efficiency of US for finding of PC might be not high, but we believe that pancreatic cancer screening with any modality (CE-CT, EUS, MRCP, and/or US) improves the prognosis of PC. Therefore, we need some evidence of the usefulness of examinations for asymptomatic persons. To address this point, we analyzed the usefulness of US for asymptomatic PC patients. Our results (good prognosis compared to symptomatic patients) might be natural as you say, but examinations that found PCs for asymptomatic individuals actually have not been performed in Japan. We are performing only surveillance for individuals at risk of PC (pancreatic cyst, family history, etc.). However, we cannot cover all PCs (<10%) with only such surveillance. Our manuscript does specifically not address the efficiency of US. We showed a better prognosis of asymptomatic PC patients by US than symptomatic patients. We hope that our manuscript supports the rationale that any modality, including US, can be recommended for pancreatic cancer screening.

Minor points
- Your English needs extensive revision by some senior authors. I listed part of the strange points below.
---->
Thank you for pointing out this error. Our English in manuscript was edited by the Elsevier editing service, however, there may still have been some odd expressions because of the poor English in our pre-edited manuscript. We apologize for our English writing. As best possible, we have revised the sentences you pointed out.

Page 1-2
The expression “there is a need identify asymptomatic patients with PC” is strange.
---->
Thank you for pointing out this error. The notion has been changed to as follows.
‘We need to identify asymptomatic patients with PC. ‘

- Page 2
The sentence “Recently, to ensure early detection of PCs, attention has been paid to patients with new-onset or rapid worsening of diabetes mellitus and surveillance for individuals with intraductal papillary mucinous neoplasm (IPMN) and family history (FH) of PC [3]. “ is strange.
---->
Thank you for your suggestion. The notion has been changed to as follows.
“Recently, to find earlier stage PCs, attention has been paid to patients with new-onset or rapid worsening of diabetes mellitus and individuals with intraductal papillary mucinous neoplasm (IPMN) and family history (FH) of PC.’’
- The sentence “These unfortunate results might be due to the fact that most patients with PC do not come from screening for IPMN and FH of PC, although these two factors are important indicators of PCs [4–7]. “ is strange.
---->
Thank you for your suggestion. The notion has been changed to as follows.
“These results might be due to the fact that most patients with PC do not come from screening for IPMN and FH of PC, although these surveillances are important means for finding of PCs [4–7]. “
- The sentence “One reason is the report that there is no evidence of cancer screening improving the disease-specific morbidity or mortality of PC [8]. “ is strange.
---->
Thanks to your suggestion, the notion has been changed to as follows.
‘’One reason is that there is no evidence of cancer screening improving the disease-specific morbidity or mortality of PC [8]’’                
- The sentence “In this study, we analyzed the prognosis of patients with PC who were divided into eight groups according to how PC was diag- nosed and evaluated the usefulness of abdominal ultrasonography (US) during medi- cal checkups of asymptomatic individuals. “ is strange.
---->
Thanks to your suggestion. I’m sorry that I have no idea to correct the sentences to better expression.
- The sentence "For diagnosis, we first used surgical specimens, and imaging studies were used if surgery was not per- formed. ” is strange.
---->
Thank you for your suggestion. The notion has been changed to as follows.
"For diagnosis, we used surgical specimens and/or imaging examinations.”
- P3
Groups 1 or 2→ group 1 or 2
---->
Thank you for pointing out this error. The notion has been changed to as follows. (See revised version). Groups 1 or 2→ group 1 or 2                
- The terms “symptomatic noticed group” and “asymptomatic noticed group “ are strange.
---->
Thank you for pointing out this error. The notion has been changed to as follows. We change that to “symptomatic group” and “asymptomatic group “                         
- P4
The expression “To further confirm the efficacy of US medical checkup compared to symptom identification” is strange.
The expression “we performed multivariate analysis of the excision ratio and prognosis of patients with were diagnosed by symptoms (242 patients) and by US medical checkup (17 patients), totaling 259 patients.” is grammatically strange.
---->
Thank you for pointing out this error. The notion has been changed to as follows.
‘’We collected the 259 patients with PC noticed by US medical checkup (17 patients) and symptoms (242 patients). We analyzed the characteristics of resected cases among the 259 patients. In addition, we performed survival analysis for these 259 patients using Cox regression hazard model.’’
 ・ The word “prescribed” is inappropriate.
---->
Thank you for pointing out this error.
We deleted ‘Prescribed’.              
- P5
The expression “compared with that had by the symptomatic noticed group “ is strange.
---->
Thank you for pointing out this error. The notion has been changed to as follows.
‘’than patients in the symptomatic group’’

- The expression “CA19-9, neutrophil to lymphocyte ratio (NLR), prognostic nutrition index (PNI) possible prognostic factors of PC were better” is strange.
---->
Thank you for your suggestion. The notion has been changed to as follows.
‘’In the asymptomatic group, the CA19-9 and neutrophil to lymphocyte ratio (NLR) were lower and the prognostic nutrition index (PNI) was higher than in the symptomatic group (median CA19-9: 104 vs. 557 U/ml, P <0.01, NLR: 2.8 vs. 3.6, P<0.001, and PNI: 48.8 vs. 46.5, P<0.01, respectively).’’

P7
The expression “In addition, the US medical checkup group had significantly lower proportion of patients with best supportive therapy than that had by the symptomatic noticed group.”
This sentence is just a paraphrasing of the previous sentence.
---->
Thanks to your suggestion. We deleted this sentence.

- P9
The univariate analysis of patients in asymptomatic noticed group plus US medical checkup group showed more resected cases in the US medical checkup group (P<0.01). What would you like to emphasize from this sentence?
---->
Thanks to your suggestion. We reconsidered and correct the sentence.
‘’The univariate analysis of patients in the asymptomatic group plus US medical checkup group showed more resected cases in the US medical checkup group (P<0.01), females (P=0.045), IPMN-derived carcinoma cases (P=0.03), patients with normal or high body mass index (BMI) (BMI ≥18.5; P<0.001), and patients with CA19-9 <425 (P<0.001).’’

- CA19-9 <425
Why did you set the cutoff value of CA19-9 at 425?
---->
Thank you for your suggestion. The value of CA19-9, 425 (U/ml) was based on the median values of patients in Groups 1, 2, and 7. We described this in the method section.

- P10
prognosis →prognoses
---->
Thank you for pointing out this error. We corrected.

- P11
Why did you set the cutoff value of BMI at 18.5.
---->
Thank you for your suggestion. The cut-off value of BMI (18.5 kg/mm2) was decided as differentiating normal weight from underweight. We described this in the method section.

- The expression  ”There were only three patients for which PC was identified at health screening centers” is strange.
---->
Thank you for your suggestion. We deleted ‘only’.

- P12
You used the sentence ”PC has the worst prognosis among all cancers, and its 5-SR is approximately 7.1% and 10% in Japan and the United States, respectively.” both in the introduction and discussion.
---->
Thank you for your suggestion. We changed the sentence. See revised version.

- The expression “Thus, overcoming the burden of PC is an urgent issue.” is strange.
---->
Thank you for your suggestion. We changed the sentence.
‘’Thus, overcoming of PC is an urgent matter.’’

- The expression “PCs that can radically cured are in Union for International Cancer Control stage 0 (in situ) and stage IA with 5-SRs of 85.8% and 68.7%, respectively [12,13]. “ is strange.
---->
Thank you for your suggestion. We changed the sentence.
“It has been increasingly recognized that the prognosis of patients with early-stage PC is favorable [11,12], and PCs that can be radically cured are stage 0 (in situ) and stage IA in Union for International Cancer Control 8th. The 5-SRs of stage 0 and stage IA are 85.8% and 68.7%, respectively [12,13].”
- The expression “Patients with stage 0 or 1A are increasing” is strange.
---->
Thank you for your suggestion. We changed the sentence.
‘’Finding of PC patients with stage 0 or IA are increasing, but their proportion in all PCs is currently still low.’’
- P13
The expression “Adju-vant chemotherapy using Tegafur Gimeracil Potassium (S-1) showed a 5-SR of 44.1% [19] and neoadjuvant chemotherapy using gemcitabine and S-1 for resected PCs also showed a 2-year overall survival rate of 63.7% [20]. “ is strange.
---->
Thank you for your suggestion. We changed the sentence. “Adjuvant chemotherapy for resected PCs using Tegafur Gimeracil Potassium (S-1) showed a 5-SR of 44.1% [19] and neoadjuvant chemotherapy for using gemcitabine plus S-1 for resectable PCs also showed a 2-year overall survival rate of 63.7% [20].’’
- The expression “as it is not required by the Ministry of Health and Welfare, Japan” is strange. 
---->
Thank you for your suggestion. We changed the sentence.
‘’In Japan, screening is recommended by the Ministry of Health and Welfare for five cancers including lung, stomach, breast, colon, and uterus neck cancers, but is not recommended for pancreatic cancer.’’             
- The expression “downregulation of mortality” is strange.
---->
Thank you for your suggestion. We corrected the phrase.
‘’downregulation -> improvement ‘’
- What’s the meaning of simpleness of US?
---->
Thank you for your suggestion.
We mean ‘’ ease of use’’. We corrected the phrase.

- The expression “prolong the prognosis” is strange.
---->
Thank you for your suggestion. We made the correction.
   ‘’prolong -> improvement’’                
- P14
What’s the meaning of "following fibrosis”?
---->
We apologize for our ambiguous expression. We corrected the phrase.
deleted ‘following fibrosis’.

- The expression "Surprisingly, 3 were alive patients and without relapse for over 7 years and had the potential of achieving complete remission. ” is strange.
---->
Thank you for your suggestion. We corrected the phrase.
‘’Of the 8 patients, 7 were alive with no relapse and with a survival time ranging from 539 days to 2,690 days at the time of this analysis. In addition, 3 of 7 were alive at over 7 years.’’
- The expression “progression of chemotherapy and surgeons’ technique “ is strange.
---->
Thank you for your suggestion. We made the correction.
‘’This shows some PCs at stage 2 have the potential of being radically cured, which could be explained by progress in chemotherapy and surgical skills.’’

Round 2
Reviewer 2 Report
Dear author
Thank you for the re-submission of your paper to our journal. As I stated in the first review, it is well known that, irrespective of the type of cancer, asymptomatic cancers are more likely to be detected in the early stages and have a better prognosis than symptomatic cancers. It, therefore, is natural that the prognosis is good if it is detected asymptomatically by some kind of image, not limited to ultrasound. However, unlike CT and MRI, ultrasonographic evaluation is greatly affected by the knowledge and experience of the doctor and technician performing the examination, as well as the physique of the patient. If you would like to emphasize the usefulness of ultrasound in mass-screening, you need to demonstrate the sensitivity and specificity of ultrasonography for identifying pancreatic lesions. It is only after this important point has been rectified that the reviewers will correct any inappropriate English expressions in your manuscript.
Author Response
Thank you for your suggestion.
As you say, the sensitivity and specificity of US for diagnosis of pancreatic cancer is a very important point.
I apologize that we could not show the detectability of pancreatic cancer with US in this study. Unfortunately, most patients were diagnosed with computed tomography and most underwent not US but endoscopic ultrasonography as additional examination. We referred to this point as a limitation of our study and discussed using past reports that showed the sensitivity and specificity of US for diagnosis of pancreatic cancer. In addition, we stated the weak points of US as you say in discussion section.
In Japan, screening is recommended by the Ministry of Health and Welfare for five cancers including lung, stomach, breast, colon, and uterus neck cancers, but is not recommended for pancreatic cancer. Thus, most health insurance associations including National Health Insurance Association are not adopting a cancer screening for PC. For the same reason, most municipals are not adopting public cancer screening for upper abdominal organs including pancreas.
We would like to emphasize the usefulness of screening for PC using any modalities such as US.
Round 3
Reviewer 2 Report
Dear author
Thank you for the re-submission of your paper to our journal. As I stated in the first review, it is well known that, irrespective of the type of cancer, asymptomatic cancers are more likely to be detected in the early stages and have a better prognosis than symptomatic cancers. It, therefore, is natural that the prognosis is good if it is detected asymptomatically by some kind of image, not limited to ultrasound. However, unlike CT and MRI, ultrasonographic evaluation is greatly affected by the knowledge and experience of the doctor and technician performing the examination, as well as the physique of the patient. If you would like to emphasize the usefulness of ultrasound in mass-screening, you need to demonstrate the sensitivity and specificity of ultrasonography for identifying pancreatic lesions. It is only after this important point has been rectified that the reviewers will correct any inappropriate English expressions in your manuscript.
Author Response
To Reviewer 2
Major point
Thank you for your suggestion. You are right.
I apologize to you for not replying to your demand.
We intend to make effort in future to analyze the detectability of PC with US and the efficacy of cancer screening for PC.